# Effects of Extrusion Pressure During 3D Printing on Viability of Human Bronchial Epithelial Cells in 3D Printed Samples

**DOI:** 10.3390/biomimetics10050297

**Published:** 2025-05-08

**Authors:** Taieba Tuba Rahman, Nathan Wood, Zhijian Pei, Hongmin Qin, Padmini Mohan

**Affiliations:** 1Department of Industrial & Systems Engineering, Texas A&M University, College Station, TX 77843, USA; zjpei@tamu.edu; 2Department of Biology, Texas A&M University, College Station, TX 77843, USA; woodn@tamu.edu (N.W.); hqin@tamu.edu (H.Q.); padminimohan@tamu.edu (P.M.)

**Keywords:** 3D printing, cell viability, extreme pressure, extrusion pressure, human bronchial epithelial cell

## Abstract

This study investigates how different levels of extrusion pressure during 3D printing affect the cell viability of human bronchial epithelial (HBE) cells embedded in printed samples. In this study, samples were printed at three levels of extrusion pressure. The cell viability was assessed through live/dead staining via microscopic imaging. The results show that increasing the extrusion pressure from 50 to 100 kPa led to a higher degree of cell death. These results demonstrate how the extrusion pressure affects the viability of HBE cells and provide a basis for future studies on pressure-induced responses in respiratory tissues.

## 1. Introduction

The human body sometimes is exposed to extreme pressures, for example, during deep-sea diving or high-altitude and aerospace operations. Extreme pressure can disrupt the homeostasis of physiological systems, resulting in a variety of adverse effects. **Hypobaric** environments, where the atmospheric pressure is lower than the normal sea-level pressure (approximately **101.3 kPa or 760 mmHg**), are typically experienced at high altitudes. Exposure to a hypobaric environment can result in high-altitude headache, high-altitude pulmonary edema or lung fluid accumulation, high-altitude cerebral edema, acute mountain sickness, and altitude bends [1,2,3,4,5,6,7]. The hypobaric environment poses a substantial risk for different populations, including aircrews and high-altitude tourists [8]. Conversely, **hyperbaric** environments occur when the atmospheric pressure exceeds the sea-level pressure, as seen in underwater diving, mining, or hyperbaric oxygen therapy. These environments can cause barotrauma [9], decompression sickness, nitrogen narcosis, tissue compression [10], cerebral arterial gas embolism, and high-pressure nervous syndrome [8,11,12]. It is important to note that decompression sickness and cerebral arterial gas embolism typically result from the rapid reduction in ambient pressure following exposure to a hyperbaric environment, rather than from the elevated pressure itself.

Table 1 summarizes several studies conducted in hypobaric and hyperbaric environments. For instance, Bao et al. investigated human physiological responses to a single deep helium–oxygen dive. They found that a single deep dive with helium–oxygen caused muscle damage, oxidative stress, and other physiological stress [13]. It is noted that most of the studies (included in Table 1) relied on in vivo models. Furthermore, the effects of extreme pressure on cellular-level responses, particularly in human bronchial epithelial (HBE) cells, remain underexplored.

Human bronchial epithelial (HBE) cells play a pivotal role in maintaining respiratory health by serving as a barrier between the external environment and the internal lung tissues. These cells are responsible for crucial functions, including maintaining airway integrity, enabling gas exchange, defending mucosa, activating the immune response, regulating fluid balance, and repairing tissue [16,17]. However, the dynamic mechanical environment of the lung, characterized by cyclic deformation during breathing, makes HBE cells particularly sensitive to mechanical stresses. Exposure to extreme pressure environments imposes mechanical and physiological challenges on these cells. Investigating the effects of extreme pressure on HBE cells is essential to understanding the cellular mechanisms underpinning these responses and developing interventions to mitigate pressure-induced respiratory damage such as pulmonary barotrauma or decompression sickness. Moreover, studying these cells under controlled conditions can inform the design of therapeutic interventions to enhance lung tissue resilience.

Some in vitro studies have suggested that mechanical stress may alter cellular morphology and disrupt the epithelial barrier, potentially impairing lung function [16,17]. However, most of the reported studies have been limited to two-dimensional cell cultures, which lack the complexity of native tissue architecture and physiological conditions of native lung tissues [16]. Given the physiological importance of HBE cells, there is a need to examine how they respond to extreme pressure within a three-dimensional (3D) model that more closely mimics in vivo conditions. Recent advances in respiratory research have led to the development of various 3D HBE models to better simulate the in vivo airway environment. One widely used approach involves culturing HBE cells at an air–liquid interface, closely mimicking native bronchial tissue. For instance, Leach et al. developed a 3D airway “organ tissue equivalent” model by UV crosslinking a hydrogel and seeding HBE cells on the apical surface, mimicking the native airway [18]. They focused on cell differentiation, barrier function, or mucus production under static or baseline conditions using 3D cell culture. They did not investigate HBE cell responses to physical stressors such as extreme pressure, which is the focus of the present study. Aiming to understand the responses of HBE cells to extreme temperature, Rahman et al. measured the cell viability of HBE cells embedded in 3D printed samples exposed to three levels of post-printing temperature using a heat block [19]. In this follow-up study, the authors evaluated the responses of HBE cells embedded in samples printed at different levels of extrusion pressure during the printing process. Another key distinction between the two studies is the printed format: cell-laden droplets were used in the previously published paper, whereas square-shaped constructs were printed in the present study. Similar studies have not been published in the literature.

The emergence of 3D bioprinting technology provides a powerful platform to study cellular responses under physiological and extreme conditions. Unlike traditional 2D cultures, 3D-printed cell-laden constructs closely mimic the native tissue microenvironment, enabling more accurate modeling of cellular behavior under mechanical stresses [20]. This approach allows researchers to replicate the architecture of the lung tissue and expose cells to controlled pressure variations, facilitating the study of viability, proliferation, and stress responses in a manner that is more representative of in vivo conditions.

This study aims to utilize 3D printing technology to create cell-laden samples of HBE cells and investigate their responses to extreme pressure. In this study, the term “extreme pressure” refers to pressure levels that significantly exceed the normal physiological range experienced by human bronchial epithelial (HBE) cells during spontaneous or mechanically assisted breathing such as ventilation. During calm, at-rest breathing, the human body generates a modest pressure swing of approximately 4 cmH_2_O (~0.39 kPa) to facilitate inhalation and exhalation. According to the literature, the diaphragm and thoracic muscles can exert maximum expiratory pressures of 60 to 120 cmH_2_O (~5.9–11.8 kPa) and maximum inspiratory pressures of negative 39 to 101 cmH_2_O (~3.9 to −9.8 kPa) [21]. During mechanical ventilation, peak inspiratory pressures (PIPs) typically are set to less than 30–35 cmH_2_O (1.5–2.9 kPa) to avoid barotrauma [22]. Additionally, coughing can transiently produce positive intrathoracic pressures up to 200 cmH_2_O (~19.6 kPa) [23]. In this study, the applied extrusion pressure, up to 100 kPa (~1020 cmH_2_O), far exceeds these physiological or clinical thresholds. By employing a 3D bioprinting approach, this study seeks to assess cell viability under three levels of extrusion pressure, offering a tool for understanding the effects of extreme pressure on lung cells.

## 2. Materials and Methods

Figure 1 illustrates an overview of the experimental design.

### 2.1. Cell Culture

Human bronchial epithelial cells (16HBE14o-, Cat. No. SCC150), were acquired from Millipore Sigma. 16HBE14o- cells were propagated as adherent culture and cryopreserved as a seed-stock system in accordance with vendor specifications. 16HBE14o- cells were then thawed and cultured in tissue culture (TC)-treated adherent culture flasks according to the procedure outlined in a previously published paper [24]. When the cells reached 90% confluency, they were then propagated further to produce enough cells for the experiments, following the procedure described in a previously published paper [24]. After cell propagation, the HBE cell concentration was approximately 1 × 10^6^ cells per milliliter of medium.

### 2.2. Preparation of Bioink

To prepare the bioink, a sodium alginate–carboxymethylcellulose (SA-CMC) solution was combined with a neutralized collagen solution. The SA-CMC solution (4% *w*/*v* of sodium alginate [A1112, Sigma-Aldrich, Saint Louis, MO, USA] and 4% *w*/*v* of CMC [A18105.36, Sigma-Aldrich, Saint Louis, MO, USA]) and the collagen stock solution (TeloCol-10, type 1 bovine collagen, 10 mg/mL, Advanced Biomatrix, Carlsbad, CA, USA) were prepared following the established protocols described in the literature [24,25]. Specifically, 1.5 mL of neutralized collagen solution was mixed with 13.5 mL of the SA-CMC solution in a sterile conical tube. The resulting bioink was stored in an ice bath prior to cell addition to prevent premature thermal gelation.

### 2.3. Preparation of Cell-Laden Bioink

To prepare cell-laden bioink, 2.2 mL of the HBE cells with medium (prepared by following the procedure described in Section 2.1) was added to the 15 mL of bioink, leading to the final cell concentration of approximately 1.3 × 10^5^ cells per milliliter of bioink.

To make it possible to assess the viability of HEB cells in 3D printed samples, 100 µL of master mix of Hoechst 33342 (Thermo Fisher H3570, Thermo Fisher Scientific, Waltham, MA, USA) and SYTOX Green (Thermo Fisher S7020, Thermo Fisher Scientific, Waltham, MA, USA) was added to the cell-laden bioink before printing the samples. The final concentration of Hoechst 33342 was approximately 8.1 µM. The final concentration of SYTOX Green was approximately 167 nM. The 30 min staining time ensured sufficient binding and signal generation without overexposure (overexposure could lead to nonspecific staining or signal degradation). Hoechst 33342 stains the nuclei of all cells, producing a highly fluorescent blue signal, detectable using a standard DAPI filter. Sytox selectively stains the nuclei of cells with compromised plasma membranes, producing a highly fluorescent green signal, detectable with a standard FITC filter.

### 2.4. Design of Printed Samples

The design, a 20 mm-by-20 mm square with a thickness of 1 mm, utilized for printing, was selected from the built-in STL design files available within the DNA Studio 4 software, integrated with the BioX6 3D printer (Cellink, BioX6, Gothenburg, Sweden). DNA Studio 4 provides a comprehensive interface that facilitates direct editing and the generation of G-code from the pre-existing design files, thereby eliminating the requirement for external slicing software or manual file importation. This sample dimension was chosen to fit comfortably within the well of a standard 6-well plate, where each well has an internal diameter of approximately 35 mm, allowing sufficient space for a printed sample, media exchange, and post-printing analysis.

### 2.5. Three-Dimensional Printing of Samples on Well Plates

In this study, triplicate samples were printed at each of the three extrusion pressure levels (50, 80, and 100 kPa). The cell-laden bioink, prepared by following the procedure described in Section 2.3, was loaded into separate cartridges for each extrusion pressure level. After printing, each well plate containing the samples was immediately placed in a humidified incubator. The levels of extrusion pressure were selected, based on preliminary printing tests, to ensure filament continuity and sample fidelity for the specific bioink concentration and nozzle size. Pressure levels outside this range could not produce comparable samples. The steps for 3D printing the samples are illustrated in Figure 2 and described below.

First, the cell-laden bioink (prepared as described in Section 2.3) was transferred to three 3 mL printing cartridges (Cartridges 1–3) from the conical tube using a sterile Luer lock syringe and adapter. Then, a 27 G nozzle was attached to each cartridge. Cartridge 1 was loaded into the pneumatic printhead of the BioX6 3D printer. Next, Well Plate 1, containing 6 wells, was placed inside the printer. At room temperature, three square-shaped samples were printed, one sample per well, at an extrusion pressure of 50 kPa. The printing parameters included a speed of 5 mm/s and an extrusion height of 2 mm. After printing, Well Plate 1 was incubated at 37 °C with 5% CO_2_ for 30 min to induce thermal gelation of the collagen-based bioink [19,24]. Finally, Cartridge 1 was unloaded from the pneumatic printhead of the printer. The same steps were repeated for Cartridge 2 (80 kPa, Well Plate 2) and Cartridge 3 (100 kPa, Well Plate 3).

### 2.6. Assessment of Cell Viability

In reported studies for 3D bioprinting, the live/dead assay method was commonly used to evaluate cell viability [19,26,27]. This method involves staining cells with a mixture of fluorophores, where one fluorophore marks all cells and the other marks dead cells, allowing for the visualization and calculation of the percentage of live cells. In this study, the cells were stained with Hoechst 33342 (8.1 µM) and SYTOX Green (167 nM) for fluorescence microscopy (details are presented in Section 2.3). An Echo revolution fluorescence microscope (model: RON-K, BICO company, Gothenburg, Sweden) was used to capture images of the samples to assess cell viability.

Well Plate 1 with 3D printed samples (prepared by following the procedure described in Section 2.5) was taken out from the incubator and placed under the microscope. All printed samples were incubated for 30 min before assessing cell viability. This incubation time was kept the same for all samples printed at different levels of extrusion pressure. Four microscopic images were randomly captured from each of the three samples. Image views were selected randomly by alternating locations in a sample to avoid any pattern or bias. Within each sample, fields of view were chosen to capture regions with evenly distributed cells, ensuring consistency in analysis. Similarly, microscopic images were captured from the samples on Well Plate 2 and Well Plate 3.

The image files in the TIFF (.tif) format were analyzed using ImageJ Fiji software (version 1.54f). For each image, the number of all cells (*t_i_)* and the number of dead cells (*d_i_*) were counted, and the cell viability (%), *V_c_*, was calculated using Equation (1) [19].(1)$$V_{c}=\frac{t_{i}-d_{i}}{t_{i}}\times 100$$

The average cell viability value of the four images from each sample was used as the data point in the statistical analysis. Therefore, there were three data points at each level of extrusion pressure.

In the previously published paper, cell viability was assessed using both microscopic imaging and plate reader analysis, with both methods yielding consistent results. Printing conditions were held constant, and droplets of uniform size and volume were printed, which allowed for accurate and comparable plate reader analysis. In contrast, in this study, the extrusion pressure was varied to print square-shaped constructs, resulting in inconsistent sample volumes at different levels of extrusion pressure. This variation could introduce confounding effects in plate reader analysis. Therefore, to ensure accurate and comparable viability analysis, only microscopic imaging was used to assess the proportion of live and dead cells in each construct.

### 2.7. Statistical Analysis

Statistical analysis was performed using OriginPro software (version 2024b). To begin, the Shapiro–Wilk test was applied to determine whether each dataset (the cell viability data across each well plate) was normally distributed. Each data point was the cell viability value for each image. If the cell viability data passed the normality test, a one-way ANOVA would be performed for the cell viability data from three well plates. If the cell viability data failed the normality test, a Nonparametric Kruskal–Wallis ANOVA would be performed. Furthermore, a mean pair-comparison test (Tukey’s post hoc comparison test) was used to analyze differences in experimental data between three experimental conditions.

## 3. Results and Discussion

The experimental data for the live cell count and dead cell count for three levels of extrusion pressure are presented in Table 2. Table 3 presents the *p*-values from the Shapiro–Wilk normality tests on cell viability data at three levels of extrusion pressure. If the *p*-value > 0.05, the test failed to reject the null hypothesis that the data follow a normal distribution [28]. Then, a one-way ANOVA was utilized to determine the statistical significance of the effects of extrusion pressure on cell viability, as the cell viability data at all three extrusion pressure levels passed the normality test. Table 4 shows the One-Way ANOVA results on the cell viability at three levels of extrusion pressure. The *p*-value for the effects of extrusion pressure on the cell viability is 0.015. Table 5 presents the *p*-values from the mean pair-comparison test for the effects of extrusion pressure on the cell viability. For a significance level of 0.05, a *p*-value less than 0.05 indicates a statistically significant difference between the means. If the *p*-value is greater than the significance level (e.g., *p* > 0.05), this means that there is no statistically significant difference between the means at the significance level of 0.05.

The effects of extrusion pressure on the cell viability are shown in Figure 3. In Figure 3, the experimental cell viability data at each level of extrusion pressure are presented as a bar chart. Each bar represents the mean cell viability data from three printed samples at each level of extrusion pressure. The error bars in this figure represent the standard deviation among the three samples at each level of extrusion pressure.

Figure 3 shows that the cell viability in the samples printed at an extrusion pressure of 50 kPa is higher than those printed at an extrusion pressure of 80 kPa and those printed at an extrusion pressure of 100 kPa. Specifically, at an extrusion pressure of 50 kPa, the cell viability is 90.03%. At an extrusion pressure of 80 kPa, the cell viability is 84.86%, slightly lower compared with 50 kPa. The cell viability at an extrusion pressure of 100 kPa is the lowest among the cell viability values at the three levels of extrusion pressure, with the value dropping to 81.01%. This suggests that a lower extrusion pressure is less damaging to the cells, as indicated by the higher cell viability values, and an extreme extrusion pressure may exacerbate mechanical stress, leading to cell membrane rupture and higher levels of cell death.

Figure 3 also shows the variation within the cell viability data, indicated by the error bars. At an extrusion pressure of 50 kPa, the standard deviation is 2.24%, indicating a moderate and consistent spread of data. At an extrusion pressure of 80 kPa, the standard deviation is 1.58%, indicating a moderate and consistent variation within the data. At an extrusion pressure of 100 kPa, the error bar is the longest, with a standard deviation of 3.54%, showing the highest level of variation. This indicates that cell viability is less consistent at higher levels of extrusion pressure. This increased spread implies that higher extrusion pressure introduces more stress, leading to more diverse cellular responses.

Figure 4 shows the fluorescence microscopy images of the samples printed at extrusion pressures of 50 kPa, 80 kPa, and 100 kPa, stained with Hoechst 33342 (blue) for all cells, and Sytox (green) for dead cells. For the samples printed at an extrusion pressure of 50 kPa, the overlay image demonstrates a healthy distribution of cells, as indicated by the prominent blue Hoechst 33342 staining. The Sytox signal in green is sparsely present, indicating minimal cell death at this extrusion pressure. The Hoechst 33342 images in Figure 4 show that the number of all cells is noticeably higher in the samples printed at an extrusion pressure of 50 kPa than those that are printed at an extrusion pressure of 80 kPa and 100 kPa. This observation suggests that a higher extrusion pressure may negatively affect cellular functions, as fewer cells were detectable by Hoechst staining. However, no direct assessment of metabolic activity was performed. This is likely because a higher extrusion pressure causes cellular damage. In other words, at higher levels of extrusion pressure, the shear stress overwhelms the cells’ ability to maintain homeostasis, and, consequently, reduces cell viability.

The results from this study are consistent with the results from most reported studies (using other cell types). The general trend in reported studies is that a higher extrusion pressure causes a lower cell viability. Table 6 presents several reported studies on the effects of extrusion pressure (mechanical stress) on cell viability in the printed samples. The cells used in these studies included HepG2 liver cells, rat adrenal medulla endothelial cells, rat heart endothelial cells, Schwann cells, 3T3 fibroblasts, hepatocarcinoma cells, human embryonic stem cells, human skin fibroblast cells, RSC96 cells, L8 cells, and HEK 293 cells [29,30,31,32,33,34,35,36,37,38]. However, there are no reported studies on how 16HBE14o- human bronchial epithelial (HBE) cells respond to extrusion pressure during bioprinting. In addition, these reported studies had contradictory conclusions. Some researchers reported that increasing the mechanical stress during 3D printing (by increasing the extrusion pressure) decreased the cell viability in the printed samples. However, some other researchers reported that variations in mechanical stress did not significantly affect the cell viability, such as a study using rat heart endothelial cells [29]. These studies, reported in Table 6, indicate that distinct cell types differ in their sensitivity and response towards extrusion pressure, and, consequently, the cell viability values differ as well.

The statistical analysis confirmed a significant difference in the viability of cells embedded in samples printed at different levels of extrusion pressure. The practical impact lies in how such reductions in cell viability could affect the functionality and reliability of 3D printed tissue models. Specifically, even a ~10% drop in cell viability in the samples printed at an extrusion pressure of 100 kPa compared with the samples printed at an extrusion pressure of 50 kPa could compromise downstream biological processes such as tissue maturation, barrier formation, and cellular signaling, especially in constructs designed for respiratory research.

The pressure-induced effects observed in this study and the temperature-induced effects reported in the authors’ previously published study [19] show distinct trends in cell viability loss. Increasing the extrusion pressure caused a gradual but statistically significant decline in viability, reflecting acute mechanical stress during the bioprinting process. In contrast, increasing the temperature caused a more pronounced reduction in viability, likely resulting from progressive thermal damage that affected cellular structures over time. These differences highlight the distinct nature of the physical stimuli: pressure imposes instantaneous mechanical forces, whereas temperature induces sustained biochemical and structural disruption.

## 4. Conclusions

This study addresses a knowledge gap in the literature regarding how the extrusion pressure during bioprinting affects the viability of human bronchial epithelial (HBE) cells embedded in printed samples. In this study, three sets of cell-laden samples were prepared using 3D printing. These samples were printed at three levels of extrusion pressure (50, 80, and 100 kPa). In the samples that were printed at an extrusion pressure of 80 kPa and 100 kPa, the cell viability was lower compared with the samples that were printed at an extrusion pressure of 50 kPa. These results provide insights for respiratory tissue engineering. The methodology developed in this study, utilizing 3D printing to create high-pressure extreme environments for cells and subsequently analyzing their responses, can be adapted for studying HBE cells under other extreme conditions. Additionally, the understanding gained from this study, regarding how the extrusion pressure affects HBE cell viability during bioprintng, can help optimize bioprinting parameters for improved cell survival and structural integrity, contributing to the development of functional respiratory tissue constructs in tissue engineering.

The range of extrusion pressures used in this study was selected based on the constraints of the BioX6 printer employed. In addition to the extrusion pressure, other variables can also affect the stress experienced by the cells embedded in the bioink during printing, for example, the bioink concentration and nozzle size. In their future studies, the authors will alter the bioink rheology and printing parameters to increase the stress level that cells experience during printing. They then will study how the composition and rheological properties of the bioink affect the behavior and viability of HBE cells in the bioink. Future studies will also investigate cellular responses to high-pressure conditions with varying exposure durations using a pressure chamber while incorporating an appropriate baseline control. Additionally, extended time durations will be incorporated to evaluate delayed effects on viability, functionality, and stress markers. Moreover, the cell viability will be assessed both in the prepared cell-laden bioink (before printing) and in the printed 3D samples over time.

In future, the scope of evaluation will be expanded to include other critical aspects, such as structural deformation, membrane permeability, and changes in gene expression profiles. In their future studies, the authors plan to include additional assays such as transepithelial electrical resistance (TEER), tight junction protein expression, and surfactant production markers, alongside a live/dead assay to better understand the cellular responses to extreme pressure exposure. Beyond studying the effects of extreme pressure, future research will explore cellular responses to other extreme environment conditions (such as hypoxia, radiation, low gravity, and dehydration).

## Figures and Tables

**Figure 1 biomimetics-10-00297-f001:**
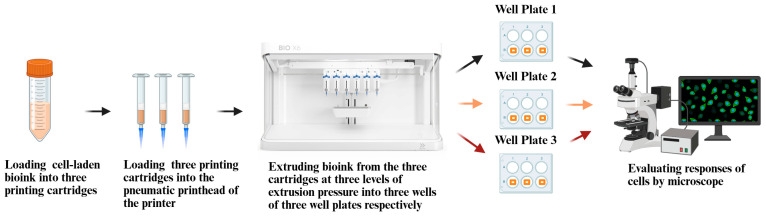
Overview of experimental design.

**Figure 2 biomimetics-10-00297-f002:**
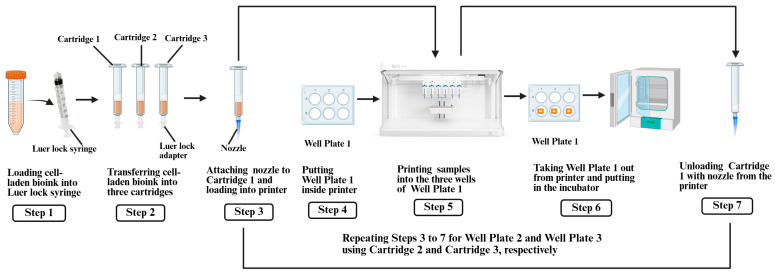
Steps for 3D printing of samples.

**Figure 3 biomimetics-10-00297-f003:**
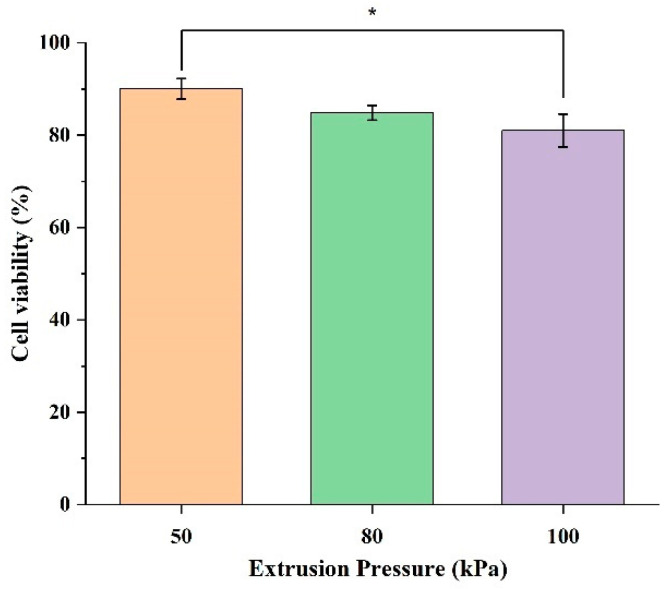
Cell viability of HBE cells encapsulated in 3D printed samples at three extrusion pressure levels (50, 80, and 100 kPa). Each bar represents the mean ± standard deviation from triplicate samples; * indicates *p*-value < 0.05.

**Figure 4 biomimetics-10-00297-f004:**
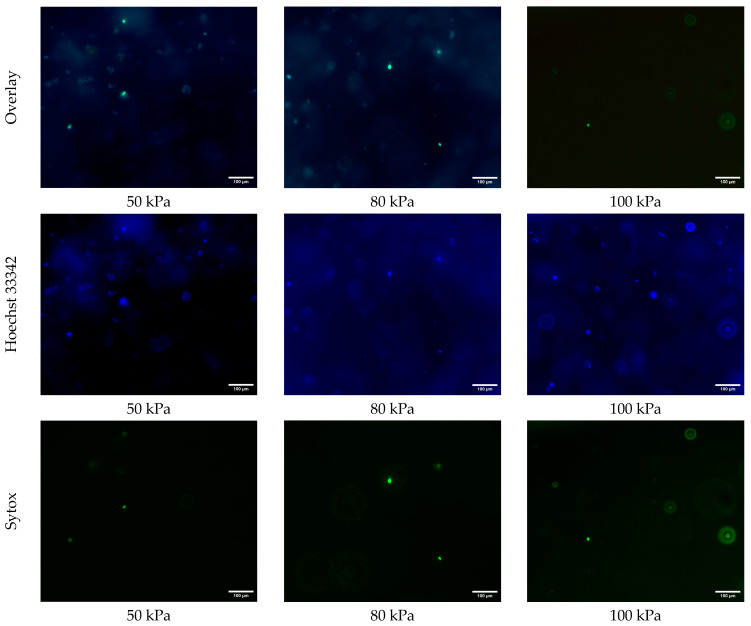
Fluorescence microscopy images of all cells (blue, Hoescht 33342) and dead cells (green, Sytox) in 3D printed samples printed at extrusion pressures of 50 kPa, 80 kPa, and 100 kPa.

**Table 1 biomimetics-10-00297-t001:** Several reported studies on extreme pressure environments.

Extreme Pressure	Test Model	Differences from This Study	Reference
Hypobaric: 5000 m above sea level	Rabbits and goats	Focused on systemic hypoxia protection using hemoglobin-based oxygen carriers; did not investigate cellular-level response in in vitro models.	[14]
Hypobaric: 5085 m above sea level	Human trekker	Investigated physiological performance and respiratory support in real-life trekking; did not investigate cellular-level response in in vitro models.	[1]
Hyperbaric: 182–200 m deep in seawater	Human divers	Analyzed microbiome adaptation in saturation divers; did not investigate cellular-level response in in vitro models.	[15]
Hyperbaric: 80 m deep in seawater	Human divers	Assessed physiological changes post-dive; did not investigate cellular-level response in in vitro models.	[13]

**Table 2 biomimetics-10-00297-t002:** Experimental data of live cell count and dead cell count for three levels of extrusion pressure.

Sample	Image	Extrusion Pressure					
		50 kPa		80 kPa		100 kPa	
		Live Cell Count	Dead Cell Count	Live Cell Count	Dead Cell Count	Live Cell Count	Dead Cell Count
1	1	9	1	5	2	14	1
	2	32	3	7	1	7	2
	3	45	2	11	1	8	2
	4	13	2	18	4	12	9
	Sample 1 mean	24.75	2	10.25	2.00	10.25	3.50
2	5	17	1	4	1	6	5
	6	25	2	8	1	10	1
	7	38	2	14	3	21	3
	8	11	2	14	1	20	1
	Sample 2 mean	22.75	1.75	10.00	1.50	14.25	2.50
3	9	11	1	3	1	24	5
	10	15	2	5	1	25	2
	11	12	3	8	1	8	1
	12	9	1	16	1	5	2
	Sample 3 mean	11.75	1.75	8.00	1	15.50	2.50

**Table 3 biomimetics-10-00297-t003:** *p*-values from Shapiro–Wilk normality tests on cell viability data at three levels of extrusion pressure.

Extrusion Pressure	*p*-Value
50 kPa	0.30
80 kPa	0.50
100 kPa	0.51

**Table 4 biomimetics-10-00297-t004:** One-way ANOVA table on cell viability data at three levels of extrusion pressure.

	Degrees of Freedom	Sum of Squares	Mean Squares	F Value	*p*-Value
Model	2	123.001	61.500	9.197	0.015
Error	6	40.124	6.687		
Total	8	163.124			

**Table 5 biomimetics-10-00297-t005:** *p*-values from mean pair-comparison tests on cell viability data between two levels of extrusion pressure.

Pair-Comparison	*p*-Value
50 kPa, 80 kPa	0.109
50 kPa, 100 kPa	0.012
80 kPa, 100 kPa	0.241

**Table 6 biomimetics-10-00297-t006:** Several reported studies on the effects of extrusion pressure on cell viability in printed samples.

Extrusion Pressure (kPa)	Bioink	Cell Type	Reference
35, 69, 138, and 276	3 *w*/*v*% alginate	HepG2 liver cells	[30]
35, 69, 138, and 276	1.5 *w*/*v*% alginate	Rat adrenal medulla endothelial cells	[31]
55 and 221	1.5 *w*/*v*% alginate	Rat heart endothelial cells	[29]
100, 200, 300, 400, and 500	6 *w*/*v*% alginate	Schwann cell, 3T3 fibroblasts	[32]
100, 200, 300, 400, and 500	10 *w*/*v*% gelatin methacrylamide	Hepatocarcinoma cells (HepG2)	[33]
50, 70, 100, and 140	1.5% *w*/*v* alginate	Human embryonic stem cells (RC-10)	[34]
21, 28, 35, 41, 48, 55, 62, 69, 103, 207, and 310	10% *w*/*v* gelatin methacrylate	Human skin fibroblast cells	[35]
50, 100, 200, and 400	2 *w*/*v*% alginate	RSC96 cells, L8 cells	[36]
55 and 83	4% Alg-4% CMC	HEK 293 cell	[37,38]

## Data Availability

The authors confirm that the data to support the findings of this study are available within the article or upon request to the corresponding author.

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
