# Peer review of "Effects of Extrusion Pressure During 3D Printing on Viability of Human Bronchial Epithelial Cells in 3D Printed Samples"

_biomimetics, 2025, doi:10.3390/biomimetics10050297_

Round 1
Reviewer 1 Report
Comments and Suggestions for Authors
The authors of the paper strive to describe the effect of bioprinting pressure on the viability of human bronchial epitheal cells. The experiment and workflow of the printing procedure is clearly explained and the effect on the cell death is clear from the presented results.
For the methodology and research carried out, it would be relevant if the authors described whether the cells are grown in suspension or as adherent. It would also be relevant to know if there was differences in the time the cells needed to wait before the cell death was analyzed. The paper should also asses the long term effects on cell death and potentially on cell functionality and/or stress responses.
The number of cells counted per image or in total for the dead cell count needs to be provided for the reader to asses reliability. Furthermore, either higher pressure or conditions producing high level of cell death should be used as positive control to ensure that the method used truly captures the cell death. Furthermore, it would be benefial for the manusript to investigate a larger pressure range or justify why the scope is limited to the current range.
Some of the conclusion in the article are not clear from the results. For example, on line 244 the authors state that low extrusion pressure preserves the membrane integrity and on line 266 that the HBE cells compromise their metabolic activity when being exposed to higher pressure. However, the manuscript includes no studies assessing the membrane integrity or the metabolic activity of the cells.
The data the authros are presenting is organized into clear tables, however, Figure 3 alone could suffice with a more extensive caption. Likewise the paragraph from lines 139 to 152, could be made more clear and consize by stating that triplicates were printed, instead of the extensive description now provided.
The introduction describes decompression sickness and cerebral artterial gas embolism as resulting from high pressure, however, these result from the decrease in the ambient pressure as the diver surfaces. It is also described in the introduction that HBE cells are particularly sensitive to mechanical stress due to their native mechanically dynamic environment. This should make the cells more adapted to mechanical stress instead.
Throughout the manuscript the term "extreme pressure" is used, however, there is no clear explanation as to what can be considered as extreme pressure for HBE cells and why.
Additionally, from reading the title, abstract and introduction, it is unclear whether different pressures were applied during printing or after printing. This should be clarified.
The authors could also improve the discussion of the results by more extensively evaluating the practical (as opposed to statistical) significance of the drop in viability.
Comments on the Quality of English Language
The english language is mostly understandable but occasionally hard to follow. There are few minor details to fix, for example: the latin terms, such as in vivo and in vitro, should be in cursive; the luer lock should be written instead of lure lock; and there is a typo on line 302.
Reviewer 2 Report
Comments and Suggestions for Authors
This paper presents a study on how high pressure affects human bronchial epithelial (HBE) cells embedded in the bioprinting samples. Here three sets of cell-laden samples were prepared by employing 3D printing. These samples were printed at three levels of extrusion pressure (50, 80, and 100 294 kPa). Basically, this study aims to understand the cell viability of human HBE cells, which were evaluated. It was observed that the samples that were printed at extrusion pressure of 80 kPa and 100 kPa, cell viability was lower compared with the samples that were printed at extrusion pressure of 50 kPa, which give important insights for respiratory tissue engineering. It was also found that the methodology developed here, by utilizing 3D printing to create high-pressure extreme environments for cells and studying their responses, can be adapted for studying HBE cells under extreme conditions. Besides that, this study presented the results on, how extrusion pressure affects HBE cell viability during bioprinting which can help to produce bioprinting parameters to improve cell survival and structural integrity, to develop respiratory in tissue engineering.
As far as my opinion concern, I found this study extremely important in HBE cell viability and tissue engineering applications. Even with the lack of computational modeling, still this work can be published as it is. Paper possesses important data, which is quite important in the field. In fact, Authors mentioned that they will do some more scientific work in their future studies

Author Response
The authors appreciate the reviewer’s positive feedback.
Reviewer 3 Report
Comments and Suggestions for Authors
The paper presents the effects of extreme pressure on human bronchial epithelial cells encapsulated in 3D printed samples. Although assessing cellular responses to extreme conditions are necessary, the manuscript must be rejected due to verbatim and near-verbatim texts similar to their previously published work in Bioengineering (https://www.mdpi.com/2306-5354/11/12/1201) with almost identical title, only this time assessing effects on extreme pressure as compared to the later's assessment of effects on extreme temperature. The previous work was not even cited in this paper even though large part of that paper was copied in this manuscript.
The paper also provided very limited assessment methods (e.i., relyong only on cell viability), when in their previous published work, they used multiple assessment methods. The paper also lacks mechanistic investigations, considering that since they already have published a similar work on the topic.
For these reasons, I strongly oppose the publication of this paper in Biomimetics.
Reviewer 4 Report
Comments and Suggestions for Authors
The article by Rahman et al investigated the viability of human bronchial epithelial cells under extreme pressure using 3D bioprinting. The idea is novel and inspiring, offering useful tools for studying the cellular environment in human respiratory system. More details about background information and data are needed.
Here are some minor suggestions for consideration:
- In the introduction on page 1 line 37, the author mentioned that most of the studies in Table 1 relied on animal models, but actually only 1 study used animal model (ref 14), while the rest 3 focused on human. Please be careful with the wording. Also, it will be useful if each of these findings could be summarized to another column in the table. The readers would then know the difference between these studies and this work.
- On page 2 line 65, the author described other 3D HBE models, what were their findings? Are these findings related to this work?
- On page 4 section 2.4, how did the author decide the printing size and volume of the cell model? What’s the printing resolution and estimated cost per assay?
- On the same page section 2.5 line 162, were the printing parameters optimized?
- On page 5 section 2.6 line 187, so the printed cells were checked right after 30 mins of incubation in the incubator. Have the author considered monitoring the cells over a longer time period like days to weeks? It would be interesting if so. Is that possible with the current bioink technologies?
- On the same page section 2.7, maybe it will be useful to check the cell viability both in the prepared cell-laden bioink (before printing) and the printed cell models, that way the viability could be normalized under each pressure condition, e.g. normalized cell viability = post-printing cell viability / pre-printing cell viability.
- For Figure 3 on page 7, instead of plotting the mean values under each pressure condition in bar plot, maybe a box plot showing the cell viability of all locations in all wells under each pressure condition will be better.
Round 2
Reviewer 3 Report
Comments and Suggestions for Authors
The authors have made substantive efforts to address my initial concerns regarding the overlap with their previous paper published in Bioengineering. They revised the title to emphasize the extreme pressure used in the printing and they also restructed parts of the abstract and the manuscript, as well as clarifying the methodology. However, the methodology still remains redundant with the previous work and the reliance solely on microscopic imaging limits the mechanistic insight into the pressure-specific cellular response. The discussion still lacks comparative analysis on how the pressure-induced effects differ from their previous paper on temperature-induced effects. Hence, I recommend reconsideration after major revisions. The authors must be able to show the analysis of the cellular efffects through other complementary assays. The paper still needs major restructuring to reduce the redundancy from the Bioengineering paper. Without these revisions, the study risks incremental contriubution despite its technically sound approach.
